# Optimal Software Versioning Strategy Considering Customization and Consumer Deliberation Behavior

**Wenjun Shu, Zhongdong Xiao \*, Ruirui Zhang and Quanyao Cao**

School of Management, Xi'an Jiaotong University, Xi'an 710000, China
* Correspondence: xzd@xjtu.edu.cn

**Abstract:** This study investigates the optimal versioning problem when a monopoly software provider bears the deliberation cost to help reduce consumer uncertainty about SaaS customization. We develop stylized models based on different production strategies and deliberation support strategies. We consider customer deliberation behavior as a new perspective on the need for a free trial. Our results indicate that a short free trial leads to free riders while a long enough free trial eliminates free riders. This is because a long free trial means that consumers easily get accustomed to the product. We also find that the seller benefits from offering deliberation support. The optimal product strategy is dependent on the deliberation support cost. When the deliberation support cost is low, the seller should provide dual products; on the contrary, the single SaaS product strategy is better with a high deliberation cost.

**Keywords:** OR in marketing; software versioning; cannibalization effect; customization; customer deliberation

## 1. Introduction

Software as a service has become an important solution for small and medium-sized enterprises to provide information services. Compared with traditional on-premises software products, SaaS products are characterized by portability, rapid implementation, and a pay-as-you-go format. According to the report, the global SaaS market share will peak at USD 716.52 billion in 2028 [1].

However, most of the operation research (OR) field, which cares about SaaS versioning, release, and channel strategies, and on-premises products, agrees that SaaS is not customizable [2–5]. In fact, as the SaaS market share increases year by year, most SaaS applications also offer customized versions. Salesforce's fully customizable CRM editions became its most popular SaaS offering. Slack company, which provides collaborative office software, offers customized SaaS products to serve various businesses. Similarly, SaaS products like RightMessage work hard on hyper-personalized experiences.

The OR field has produced the literature on SaaS customization and studies on the impact of SaaS customization on the software market structure when software enterprises stay in transformation period [5]. This study points out that when SaaS customization efficiency is low, on-premises products provided by monopolistic software vendors dominate the market; when the degree of SaaS customization is not low, the customized SaaS competes with on-premises products [5]. However, the previous study does not show the impact of the free trial version of SaaS customization on the market structure.

When we consider the impact of SaaS customization on the seller's product versioning strategies, we cannot ignore the uncertainty of consumers' evaluation of emerging SaaS customization products. Therefore, we consider the research of SaaS customization and consumer deliberation behavior on the software provider's version strategy. Consumer deliberation refers to the processes and activities of releasing consumers' preference for products and determination of their willingness to purchase [6]. In order to reduce consumers' uncertainty in evaluating new products, sellers often provide some support (such

as store displays and sales assistants) to encourage consumers to learn their own preferences [6,7]. Zeithaml [8] proposes that consumers can evaluate the attributes of a product only after experiencing it. This article focuses on the software industry, and how a free trial of customized SaaS products can essentially be regarded as a commercial practice of the seller's deliberation support. For example, SAP Business One is an ERP suite for small and medium-sized companies that can be deployed on-site or in the cloud (SaaS). It can be customized for related industries or businesses, and it also supports free trials to assist consumer deliberation.

However, most previous studies only considered the impact of consumer deliberation cost on the seller's decision making [6,7,9] while ignoring the seller's deliberation support cost. This paper considers the cost of seller deliberation support, which distinguishes it from previous studies on deliberation behavior. When a seller provides SaaS customizable services, how does the seller's deliberation support affect consumer decision making if the seller provides deliberation support? Should the monopoly software vendor provide deliberation support? What is the optimal product strategy for sellers considering a custom SaaS free trial?

We designed four models to investigate the versioning problem and deliberation support strategies: (a) Single-product strategy Without Deliberation support (SWOD); (b) Dual-product strategy Without Deliberation support (DWOD); (c) Single-product strategy With Deliberation support (SWD); (d) Dual-product strategy With Deliberation support (DWD). By comparing different strategies, we attempt to answer the proposed research questions. We have found that it is more beneficial to provide deliberation support for custom SaaS products when the seller chooses a single SaaS product strategy. When a seller chooses a dual-product strategy, a choice of whether to provide deliberation support for custom SaaS products depends on the seller's deliberation support cost. Interestingly, when the seller offers shorter deliberation times, free riders are generated in the customized SaaS market; however, these free riders would disappear over a long enough deliberation period. Finally, we reveal that when the seller has a low deliberation support cost, DWD is the best strategy; when the cost of deliberation support is high, SWD is the optimal strategy.

## 2. Relevant Literature

Several branches of research are related to this work: the software versioning strategy, including the SaaS business model and traditional software, customer deliberation, and the software free trial research.

### 2.1. The Free Trial of Experience Products

Abundant studies in the operational research field focus on free trial strategy, especially in software versioning and online service systems. Hua et al. [10] claim that the optimal length of a free trial in an online service market is not more than 1/3 of the lifespan of the initial version. They catch the upgrade feature of online services and study a two-stage problem from the perspective of microeconomics. Wang and Sun [11] consider boundedly rational consumers who make purchase decisions after experiencing free trials. They reveal the optimal service pricing and service capacity. Yoganarasimhan et al. [12] empirically found that the outcome of a 7-day free SaaS trial is better than that of a uniform 30-day free trial.

From the perspective of the free trial strategy, many researchers consider network or externality effects in free trials. Software providers use free trials or sample versions to resolve the customer's uncertainty problem [2,4,13,14]. In a monopoly setting, Cheng et al. [2] think the usage of the free trial can increase the installed user base and study the tradeoffs between the positive network effect and negative externality effect. They think whether to provide the free trial depends on network intensity. Cheng and Liu [15] also focus on the tradeoffs between reduced uncertainty and demand cannibalization. They work on the optimal free time based on consumer learning behavior. Yi et.al [16] have found that in a reselling structure, the usage of the free trial can be unexpectedly used to fight against double marginalization.

However, Nan et al. [17] have found that in the competition scenario, the network effect is negative, which is in contrast with the results of monopoly settings. Liu et al. [14] have researched how customers' prior beliefs plays a key role when software providers decide on a free trial strategy in duopoly and monopoly settings. They demonstrate that it is better to offer a free trial when the quality of substituting product is high. Wu et al. [18] examine the tradeoff between the negative effects and positive effects of a free trial in an oligopoly market. They claim that although the free trial resolves consumer uncertainty, it also promotes consumer switch because of poor fits after free trials.

Liu et al. [14] make comparisons between advertising and free trials. Nan et al. [17] also take the seeding strategy into consideration when choosing the promotional program and finding suitable situations for each strategy. In addition, Liu and Li [13] compare the coupon and free trial scenarios for cloud computing promotion. They give suggestions to cloud computing providers on when to choose the targeted coupon.

This work is different from the previous study. We study the optimal free trial time from the perspective of consumer deliberation behavior. Since research has shown that the consumer learning effect can affect the optimal free trial length [10], it has not provided suggestions on when to offer a free trial. In this study, we not only study the optimal free trial length but also give advice on when to conduct free trial promotion programs.

### 2.2. Customer Deliberation Behavior

Researchers in marketing raise many opinions on consumer uncertainty, especially valuation uncertainty. Consumers always bear cost of finding out their true preferences and quality valuation. Guo and Zhang [6] empirically came up with the theory of contextual deliberation, which may explain the customer preference construction and customer irrational behavior. Guo and Zhang [6] define customer deliberation as a preference-learning activity, and it is costly for customers to find their preferences. The customer who deliberates may have a high or low valuation of the quality or may have a homogeneous valuation of the quality [6,9]. Guo and Zhang [6] have interesting findings that when the price of a high-end product is reduced, customer deliberation can be induced, and low-level quality can prevent deliberation when the price is low. However, Guo and Wu [19] show that a high-quality firm induces deliberation with high pricing whereas a low-quality firm prevents deliberation by setting low prices. Xiong and Chen [9] use consumer deliberation to design a tailored product line, and the seller pays for the customers' deliberation cost. Xu and Zhou [20] take consumer deliberation into consideration and find conditions when commonality surprisingly reduces the cannibalization in the product line. Following Xiong and Chen [9,21], we consider the residual valuation uncertainty, which means that consumers should have bought the product without deliberation but have left with nothing. They also find situations where sellers should abandon seller-induced learning. Li et al. [4] reveal the impact of deliberation cost in a decentralized supply chain and find that a lower wholesale price leads to a lower retail price when deliberation cost is high.

Previous research on customer deliberation behavior has always considered quality valuation as a heterogeneous value to match the heterogeneous quality and price discrimination. This work is different from those because it fills in the gap when the customer valuation is a continuous distribution.

Compared with the previous studies, we have several differences. First, inspired by Xiong and Chen [21] who came up with consumer-induced learning, we consider seller deliberation support cost while the previous studies only look at customer deliberation cost. We claim that whether the seller provides deliberation support depends on the deliberation support cost. Second, we link the deliberation behavior to the software free trial strategy. From the perspective of consumer deliberation behavior, this work offers a free trial strategy for newly customized SaaS. Third, there are few studies in the operational research field that examine the SaaS customization. This work enriches the versioning strategy when SaaS is customized.

The rest of this work is organized as follows. In Section 3, we show the model setup and derive the optimal solutions for each strategy. We draw results in Section 4 and discuss how to support customer deliberation and what the optimal product strategy is. We develop a discussion in Section 5 which compares and contrasts solutions/results presented in this study with the existing work. Finally, we draw conclusions in Section 6.

## 3. The Model Methods

Unlike previous studies that regarded SaaS products as standard versions (which cannot be customized), this paper examines the version strategy involved in consumer deliberation behavior and customized SaaS products. We consider that monopoly software vendor sells customized SaaS and traditional on-premises products directly to customers. Following the previous literature, we assume that a consumer's evaluation of traditional on-premises products is $v$, and $v$ is uniformly distributed at [0,1]. While consumers recognize customized SaaS products as $kv$, consumers rationally choose customized SaaS and on-premises products. According to Basu and Bhaskaran [22] there are three categories of customization, and on-premises software is more similar to the "job-shop" approach, which is customized according to customer requirements. SaaS products, on the other hand, are a type of mass customization, where consumers choose customization from preset options. The third one is co-designed. In customized SaaS products, the higher the degree of customization, the more difficult the scale effect is [23]. In general, compared with traditional on-premises customization products, SaaS products are less customized. In this way we consider $k < 1$. Consumers' customization efforts for SaaS products are $r_s$, and the customization efforts for traditional products are $r_p$. We assume $r_s < r_p$. Consumer deliberation costs are homogenous and denoted as $l$, the net utility of consumers buying custom SaaS products (new entrant) is $u_s = kvq - p_s - r_s - l$ and the utility of purchasing on-premises products (incumbent) is $u_p = vq - p_p - r_p$, where $p_s$ and $p_p$ are the price of customized SaaS products and on-premises products, respectively, and the product quality is $q$. The customers make rational choices, and they choose the SaaS product when $u_s > 0$ (single-product strategy) or $\{u_s > u_p, u_s > 0\}$ (dual-product strategy).

Providing free trials of customized SaaS products can be regarded as deliberation support activities provided by sellers. According to the assumptions of Cheng and Liu [15], we believe that the WTP (willingness-to-pay) of consumers who use g free trial products increases linearly over time, and we denote the increasement as $\delta\tau$. After consumers experience the free version, they may increase or decrease their product ratings. To simplify the model, we assume that the average consumer learning speed is $\delta$. When $\delta < 0$, seller offering deliberation support is very unsatisfactory because of the reduced willingness-to-pay (WTP) of consumers. Therefore, we discuss a case of average learning rates $\delta > 0$ [15,24]. The utility of consumers to purchase customized SaaS products (new entrants) increases with the amount of free time, denoted as $u_s = kq(v + \tau\delta) - p_s - r_s - l$, and the utility of purchasing on-premises products is $u_p = qv - p_p - r_p$.

As a new entry into the market for customized SaaS products, consumers want to get access to product information to understand their preferences and evaluations of products. Sellers carry out deliberation campaigns to support consumers' cognition of new products, such as launching free trials. Different from previous studies that focused only on consumer deliberation cost, we consider sellers' deliberation campaign costs to support consumers' deliberation behavior. In addition, the seller is continually providing maintenance to their free-trial software. In this paper, the cost of seller deliberation activity is correlated with deliberation time $\tau$, denoted as $h\tau$, where $h$ is the deliberation support cost per time. Deliberation support cost is involved with the seller's promotion cost, management cost, infrastructure maintenance cost, and other variable costs over time.

The seller has two product strategies, a single-custom SaaS (or on-premises) strategy and a dual-product strategy. One of the characteristics of software products is that there are no marginal costs [4]. This article only considers the initial development cost of software products and does not consider the update iteration of software quality. Thus, referring to

Choudhary [25], this paper considers the fixed-cost function as a linear function (according to IS practice, the product updating is considered a fixed-cost, which is a concave function). We assume that the initial development cost function under the single-product strategy is a linear function of quality, denoted as $cq$, and the initial development cost function under the dual-product strategy is denoted as $dq$, where $c > d$.

When the seller does carry out the deliberation campaign, the seller's profit function under the single-product strategy is $\Pi = px - cq$, and the profit function under the dual-product strategy is $\Pi = p_s x_s + p_p x_p - dq$. When the seller's deliberation campaign is conducted, the profit function under the single-product strategy is $\Pi = px - cq - h\tau$, and the profit function under the dual-product strategy is $\Pi = p_s x_s + p_p x_p - dq - h\tau$.

This paper examines four models to reveal the impact of seller-provided deliberation on product lines and the effect of seller' deliberation cost on their decision making. The subscripts in the model represent each of the four base models. In the subscript, $j = \{s, p\}$ represents SaaS products or on-premises products.

### 3.1. SWOD Strategy

In this scenario, the software vendor provides a single product with no deliberation support. The consumer utility of purchasing a customized SaaS product (subscript $s$) is $u_{1s} = kvq_{1s} - p_{1s} - r_s - l$, and the consumer net utility of purchasing an on-premises (subscript $p$) product is $u_{1p} = vq_{1s} - p_{1p} - r_p$. The profit functions of the seller under the single SaaS product strategy and the single on-premises product strategy are $\Pi_{1s} = p_{1s}x_{1s} - cq_{1s}$ and $\Pi_{1p} = p_{1p}x_{1p} - cq_{1s}$, respectively. After a simple derivation, we make $d_1 = \sqrt{-(4ck - k^2)^{-1}}$ and $d_2 = \sqrt{1 - 4c}$. The price, quality, and market demand of customized SaaS products and on-premises products are as follows:

(a) The customized SaaS products: $p_{1s} = \frac{(kd_1 - 1)(r_s + l)}{2}$; $q_{1s} = d_1(r_s + l)$; $x_{1s} = \frac{1}{2} - \frac{1}{2kd_1}$.

(b) The on-premises products: $p_{1p} = \frac{(1 - d_2)r_p}{2d_2}$; $q_{1p} = r_p - \frac{(d_2 - 1)r_p}{d_2}$; $x_{1p} = \frac{1}{2} - \frac{d_2}{2}$.

Substituting the above formulas into the profit function of the seller, we obtain that the profit of the seller under the single SaaS product and the single on-premises product is $\Pi_{1s} = \frac{(r_s + l)(1 - kd_1)}{2kd_1}$ and $\Pi_{1p} = -\frac{r_p(4c - 1 + d_2)}{2d_2}$, respectively.

We find that under the single-product strategy, the greater the consumer's effort in product customization, the higher the product quality, i.e., $\frac{\delta q_{1s}}{\delta r_s} > 0$, $\frac{\delta q_{1p}}{\delta r_p} > 0$. Thus, the higher the customization effort, the higher the product price, i.e., $\frac{\delta p_{1s}}{\delta r_s} > 0$, $\frac{\delta p_{1p}}{\delta r_p} > 0$. However, the seller needs to pay higher cost for the increased product quality. The higher the consumer customization effort, the lower the seller's profit, i.e., $\frac{\delta \Pi_{1p}}{\delta r_p} < 0$, $\frac{\delta \Pi_{1s}}{\delta r_s} < 0$.

When SaaS customization has higher purchasing effort, i.e., $r_s + l > \frac{kd_1(4c - 1 + d_2)r_p}{k(d_1 - 1)d_2}$, we also find that the single on-premises products are more profitable ($\Pi_{1s} < \Pi_{1p}$). When consumers' investment in SaaS customization is higher than the threshold (i.e., $r_s \geq \frac{r_p}{d_1 d_2}$), the quality of SaaS products may be better than that of on-premises products. However, sellers need to pay substantial costs for high-quality customized SaaS, resulting in a lower profit of the single SaaS product than that of the single on-premises product.

### 3.2. DWOD Strategy

In this scenario, the software vendor offers a dual product and no deliberation support activities. Consumers choose rationally from SaaS customization products or on-premises products. The consumer's net utility for purchasing a customized SaaS product is $u_{2s} = kvq_{2s} - p_{2s} - r_s - l$, the utility function for purchasing an on-premises product is $u_{2p} = vq_{2s} - p_{2p} - r_p$, and the profit function of the seller is $\Pi_2 = p_{2s}x_{2s} + p_{2p}x_{2p} - dq_2$.

After a simple derivation, we get the pricing and quality decisions and profits under the dual-product strategy without deliberation:

$$p_{2p} = \frac{(-4d+1)(r_p k^2 - r_p k) + d_3}{4k(k-1)(4d-1)}; \quad p_{2s} = \frac{d_3}{(8d-2)(k-1)} - \frac{r_s + l}{2}$$

$$q_2 = \frac{d_3}{k(4d-1)(k-1)}; \quad \Pi_2 = \frac{(4d-1)\left(-k r_p^2 + 2k(r_s+l)r_p - (l+r_s)^2\right)}{2d_3} - \frac{r_p}{2}$$

$$d_3 = \sqrt{-8k(k-1)\left(rp(l-rp/2+rs)k - 1/2(1+rs)^2\right)(d-1/4)}$$

In the dual-product strategy, when the total effort required to purchase a customized SaaS product is higher than the customization effort of an on-premises product ($r_s + l > r_p$), some consumers of SaaS products turn to on-premises products, and the dual-product strategy degenerates into the single-product strategy. We assume $r_p > r_s + l$, indicating that the dual-product strategy makes sense at this time.

**Proposition 1.** *When both products have no deliberation support, (a) In the dual-product strategy, the software vendor's profit decreases monotonically on $r_s$, i.e., $\frac{\delta \Pi_2}{\delta r_s} < 0$. (b) Whether it is a single-product strategy or a dual-product strategy, the profit decreases monotonically on $l$, i.e., $\frac{\delta \Pi_{1s}}{\delta l} < 0$, $\frac{\delta \Pi_2}{\delta l} < 0$.*

Proposition 1 (a) shows that the seller's profit decreases on $r_s$ in the dual-product strategy. We find that higher customization effort results in lower sales of dual products and higher price. However, the increased price cannot make up for the loss in sales, so the higher the SaaS customization effort, the lower the profit of the dual-product strategy.

With the increase in price and product quality, sellers need to pay more. The profit brought by the price increase is inferior to the loss brought by the cost increase. When the deliberation cost is higher, the seller's profit is lower. Whether it is the single-product strategy or the dual-product strategy, we observe that profits decrease with increasing consumer deliberation costs, i.e., $\frac{\delta \Pi_{1s}}{\delta l} < 0$, $\frac{\delta \Pi_2}{\delta l} < 0$.

The existing literature points out that consumers show deliberation behavior when the product price is higher (Li et al. [7]). However, our study finds that when consumers are willing to pay a higher deliberation cost, the product can often obtain a higher price. After all, it is the customers who pay for whatever the seller provides.

*3.3. SWD Strategy*

Proposition 1 shows that the higher the consumer's deliberation cost, the lower the seller's profit is. Li et al. [7] find that retailers carrying out consumer empowerment activities can reduce consumer deliberation costs and promote channel profits. In Scenario 3.3, we consider whether the seller's deliberation support can improve the seller's profit. Different from Li et al. [7], the seller's deliberation support in this paper does not directly reduce the consumer's deliberation cost but increases the consumer's valuation. We believe that a free trial version can fulfill some of the consumer's needs or solve some of the consumer's problems. Consumers can remove uncertainty about new products from deliberation support activities [21] and increase product evaluations [15]. In this scenario, the seller provides deliberation support for the single SaaS product strategy, while the traditional single on-premises product strategy does not provide deliberation support.

Note that $d_4 = \sqrt{(\delta^2 k^2 \tau^2 + 2\delta k^2 \tau - 4ck + k^2)^{-1}}$, we calculate that the optimal pricing under the single SaaS product strategy is $p_{3s} = \frac{(r_s+l)(kd_4(\tau\delta+1)-1)}{2}$, the optimal product quality is $q_{3s} = (r_s + l)d_4$, and seller's optimal profit is $\Pi_{3s} = \frac{(r_s+l)}{2d_4 k} - \frac{(1+\delta\tau)(r_s+l)}{2} - h\tau$. The results and analysis of the single on-premises product strategy are the same as for the SWOD strategy.

Likewise, we find that product price and quality increase with consumer customization efforts and deliberation efforts, and seller's profits decrease in consumer deliberation efforts.

**Proposition 2.** *The optimal deliberation time of optimal single SaaS product strategy is* $\tau^* = -\frac{1}{\delta} + \frac{\delta(r_s+l)+2h}{\delta}\sqrt{\frac{c}{kh(\delta(r_s+l)+h)}}$. *When $\tau < \tau^*$, the seller's profit increases when the deliberation support time and the deliberation cost increase $\frac{\delta^2\Pi_{3s}}{\delta\tau\delta l} > 0$.*

When the seller provides deliberation support for a single customized SaaS product strategy, the fulfilled requirements are $x_{3s} = \frac{1}{2} + \frac{\tau\delta}{2} - \frac{1}{2kd_4}$, which increase with deliberation time. When the deliberation time increases, both the price and quality of SaaS increase. When $\tau$ is small, the profit brought by increase in SaaS prices and demand dominates the profit of the seller; when $\tau$ is large, the high costs due to increased quality of SaaS products and deliberation support make for the profit loss of the seller, and the profit of the SaaS product strategy decreases in $\tau$. We find the optimal deliberation support time is $\tau^* = -\frac{1}{\delta} + \frac{\delta(r_s+l)+2h}{\delta}\sqrt{\frac{c}{kh(\delta(r_s+l)+h)}}$, which is the optimal free time.

In the previous part, we mentioned that the seller's profit decreases with the increase of the consumer's deliberation cost, i.e., $\frac{\delta\Pi_{3s}}{\delta l} < 0$. But the seller's profit increases with the increase of the deliberation support time, i.e., $\frac{\delta\Pi_{3s}}{\delta\tau} > 0$ (when $\tau < \tau^*$). Then, we find that the seller's profit increases when the deliberation support time and the deliberation cost increase $\frac{\delta^2\Pi_{3s}}{\delta\tau\delta l} > 0$. This shows that it is beneficial to the seller to carry out the deliberation activity for a certain period of time.

### 3.4. DWD Strategy

In this scenario, sellers offer both customized SaaS and on-premises products and carry out deliberation support activities for SaaS products. At this time, the utility function of consumers purchasing SaaS products is $u_{4s} = kq(v + \delta\tau) - p_{4s} - r_s - l$, the utility function of purchasing on-premises products is $u_{4p} = qv - p_{4p} - r_p$, and seller's profit function is $\Pi_4 = p_{4s}x_{4s} + p_{4p}x_{4p} - dq - h\tau$.

We let $d_6 = \sqrt{-\left(r_p(2l - r_p + 2r_s)k - (l + r_s)^2\right)k((\tau^2\delta^2 + 4d - 1)k - 4d + 1)}$.

The optimal solutions for this scenario are $p_{4p} = \frac{d_6}{(2\tau^2\delta^2+8d-2)k^2+(-8d+2)k} - \frac{r_p}{2}$, $p_{4s} = \frac{d_6(1+\delta\tau)}{(2\tau^2\delta^2+8d-2)k-8d+2} - \frac{r_s+l}{2}$, and $q_4 = \frac{2d_6}{k(\delta^2k\tau^2+4dk-4d-k+1)}$. Seller's optimal profit is $\frac{(k\delta^2\tau^2+(4d-1)(k-1))(r_p(2l-r_p+2r_s)k-(l+r_s)^2)}{d_6(2k-2)} + \frac{2h\tau+r_p}{2} - \frac{\delta k\tau r_p}{2k-2}$.

In the dual-product strategy, we assume $kr_p > r_s + l$ to make $x_{4s} > 0$.

**Proposition 3.** *When the deliberation support cost of the seller is low, i.e., $h < \frac{(r_pk-\phi)\delta}{2-2k}$, the seller's profit increases monotonically in the deliberation time $\tau$, i.e., $\frac{\delta\Pi_4}{\delta\tau} > 0$; when $h > \frac{(r_pk-\phi)\delta}{2-2k}$, if $\tau > \tau_\Pi$, then $\frac{\delta\Pi_4}{\delta\tau} > 0$.*

From Proposition 3, we find that the selling price of the SaaS product decreases over the deliberation time $\tau$ when the seller's cost of providing deliberation support is higher, i.e., $h > \frac{(r_pk-\phi)\delta}{2-2k}$ ($\tau > \tau_\Pi$, $\tau_\Pi$ see Appendix A). More and more consumers benefit from the deliberation support activities, and the purchase of SaaS customization increases, which makes on-premises products less expensive and total demand increases with deliberation time. Increased sales dominate increased deliberation support costs. Conversely, sellers' profits increase in deliberation time $\tau$ when deliberation support costs are lower. When the maintenance cost is lower, the seller's profit always increases in $\tau$. That is because the added benefit of providing deliberation support always covers the cost of providing deliberation support. In general, deliberation support for seller is beneficial if there is sufficient deliberation time.

## 4. Results

In this section we show the model results and answer the research questions.

### 4.1. How Does Seller Deliberation Support Time Affect Purchase Decisions?

First, we clarify the impact of seller' SaaS deliberation support time on the needs of the single SaaS product. We find that the undifferentiated consumption point of SaaS customized products in SWOD case is $v_{1s} = \frac{kd_1 + 1}{2kd_1} = \frac{1}{2} + \frac{1}{2kd_1}$. In SWD case, we have indistinguishable consumers who buy the SaaS product and those who do not buy the SaaS product at $v_{3s} = \frac{1}{2kd_4} - \frac{\delta\tau - 1}{2} = \frac{1}{2} + \frac{1}{2kd_4} - \frac{\delta\tau}{2}$. We find that the market structure of consumers for the single customized SaaS product strategy varies over the time of seller deliberation support.

**Proposition 4.** *The short-term deliberation support of seller will result in free riders generated by the single SaaS product strategy.*

As can be seen from Figure 1, a short period of deliberation produces free riders, while a long deliberation will prevent free riders from being generated. When the seller has no deliberation support, the users in SWOD strategy (the single SaaS product) are distributed in the area of Figure 1① (in the thick dotted line frame), where $v_2 = v_{1s}$. For SWD strategy, when the deliberation support time is short, i.e., $\tau < \tau_1 = \frac{d_1 - d_4}{d_1 d_4 \delta k}$, SaaS consumers are located in the area of Figure 1③ (shaded line), where $v_3 = v_{3s}$ (where $v_{3s} > v_{1s}$). We observe that under the deliberation strategy, free riders are generated in a shorter deliberation time. The consumers of region ② are the free riders generated by the deliberation support.

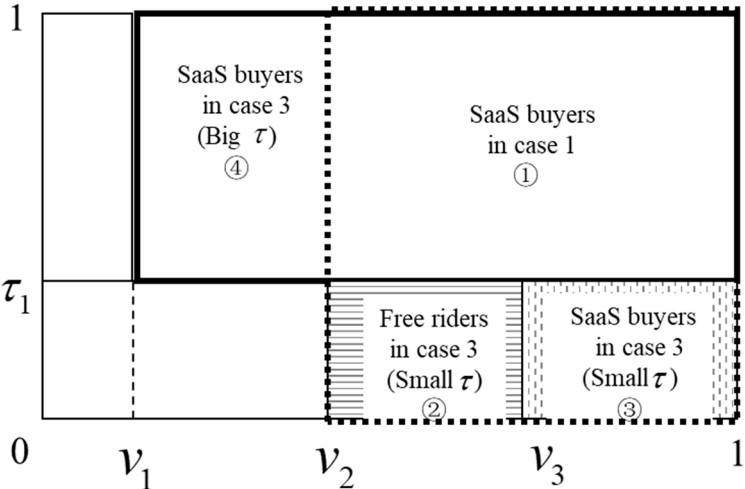

**Figure 1.** The distribution of users under the single SaaS product strategy.

When the seller' deliberation support time is longer, i.e., $\tau > \tau_1$, we have $v_{3s} < v_{1s}$. Let $v_1 = v_{3s}$ in Figure 1. At this time, the SaaS consumers of SWD case are located in the area of Figure 1④ (thick solid line box). In the no-competition context, customizable SaaS consumers have a habit of using the product for a long time, which make it difficult to switch. Therefore, the extended trial period motivates the consumer to purchase.

Next, we observe the effect of seller deliberation support time on demand in the dual-product strategy. In DWOD scenario, the indifferent consumers who buy SaaS products and those who do not buy SaaS are located in $v_{2s} = \frac{(4d - 1)(l + r_s)(k - 1)}{2d_3} + \frac{1}{2}$, and the indifferent consumers who buy SaaS products and on-premises products are located in $v_{2p} = \frac{(l - r_p + r_s)(4d - 1)k}{2d_3} + \frac{1}{2}$. In DWD scenario, indifferent consumers of customized SaaS products and on-premises products are located at $v_{4p} = \frac{-\delta k\tau + k - 1}{2k - 2} + \frac{\left((\tau^2\delta^2 + 4d - 1)k - 4d + 1\right)(l - r_p + r_s)k}{d_6(2k - 2)}$,

and consumers who purchase SaaS customizations and those who do not buy any products are located at $v_{4s} = \frac{\delta^2 k\tau^2 + (4d-1)(k-1)(l+r_s)}{2d_6} + \frac{1-\delta\tau}{2}$. Next, we show the customer divisions by using these indifference curves.

**Proposition 5.** *In the dual-product strategy, when the deliberation support time is longer, the number of SaaS consumers increases while the number of the on-premises-product consumers decreases. Seller' deliberation support increases the market share of SaaS products.*

The distribution of consumers involved in the dual-product strategy is shown in Figure 2: two straight lines parallel to the *X*-axis divide the consumers of on-premises products (rectangular area III), SaaS product consumers (rectangular area II), and consumers who do not purchase any products (rectangular area I). When the seller provides deliberation support for the dual-product strategy, area ② shows new potential SaaS consumers, who initially buy nothing. Area ① shows the cannibalization between SaaS and on-premises products. It indicates that new SaaS consumer group has been transferred from the on-premises market under the support of seller' deliberation. When the seller carries on deliberation support for the dual-product strategy, the SaaS consumer market structure changes from the original rectangular area II to the irregular area ① + II + ②.

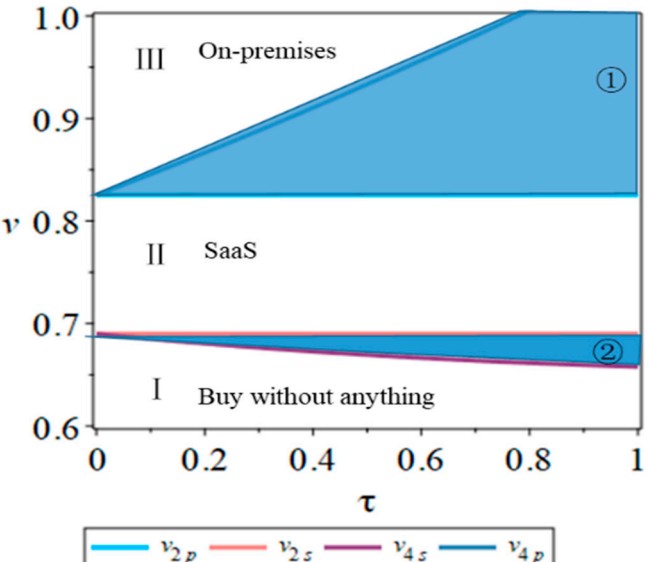

**Figure 2.** The distribution of users under the dual-product strategy ($\delta = 0.1, k = 0.8; d = 0.2; c = 0.1; r_p = 0.1; r_s = 0.05; l = 0.02$).

It can be seen from the reduced area I and area III that the seller' deliberation support for customized SaaS will reduce the number of users who buy without anything, and will also erode the sales of on-premises products.

### 4.2. Should the Seller Provide Deliberation Support and When?

First, we use numerical simulations to investigate the impact of seller's deliberation support on on-premises product profits. We set the parameters of the numerical simulation as $\delta = 0.1, k = 0.8; d = 0.2; c = 0.1; r_p = 0.1; r_s = 0.05; l = 0.02$. The parameter settings are as follows: (1) It should make a dual-products strategy exist. (2) It should make the result more intuitive. (3) It can be repeated. We find that the demand for on-premises products in DWD strategy is lower than that in DWOD strategy (Figure 3a). After the seller provides deliberation support (for SaaS products), the price of on-premises products decreases (Figure 3b blue line) and revenue decreases (Figure 3c blue line).

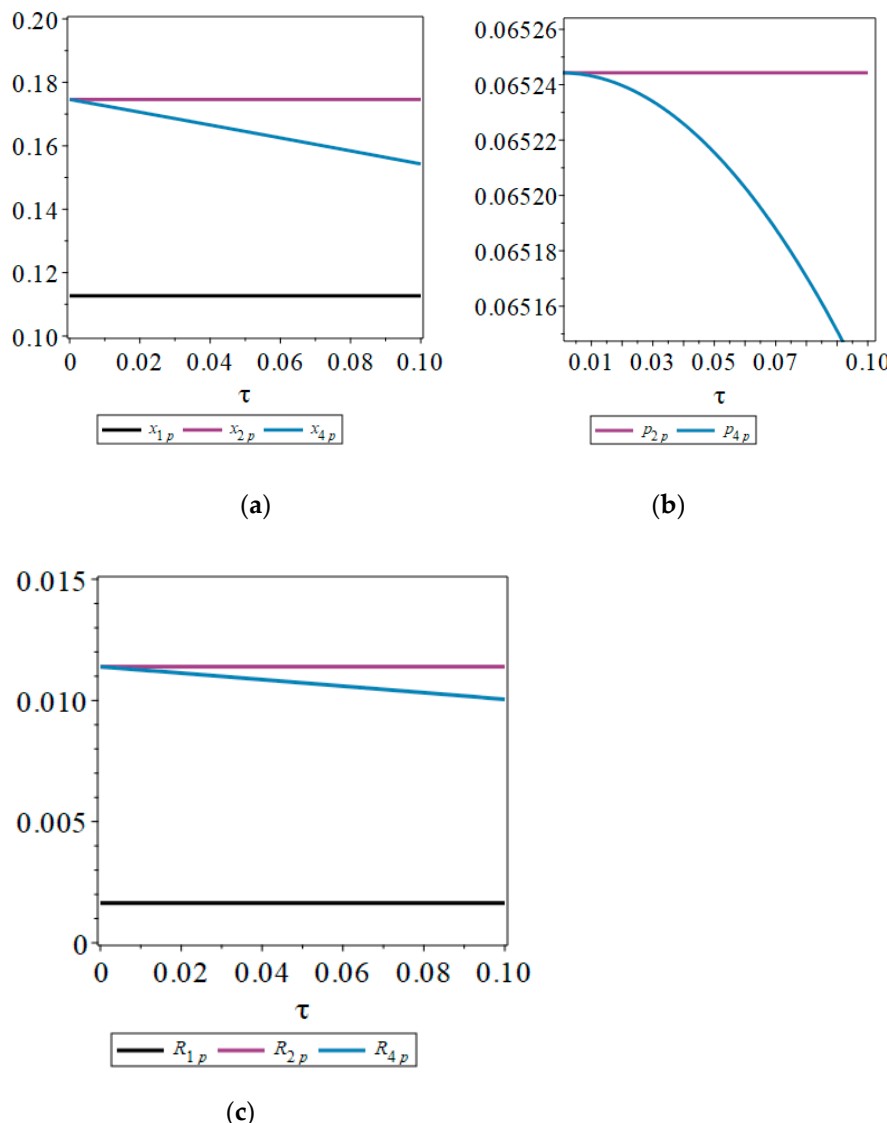

**Figure 3.** The comparison between different strategies: (**a**) the demands (**b**) the prices. (**c**) the revenues.

Through numerical simulation, we reveal that the SaaS customized deliberation support may hurt the sales and profits of on-premises products.

Next, we reveal the impact of deliberation support on product strategies in decision-making in Proposition 6.

**Proposition 6.** *For the single SaaS product strategy, we have* $\Pi_{3s} > \Pi_{1s}$*; for the dual-product strategy, when* $h < \frac{(r_p k - \phi)\delta}{2 - 2k}$*, we get* $\Pi_4 > \Pi_2$*, but when* $h > \frac{(r_p k - \phi)\delta}{2 - 2k}$*, we have* $\Pi_4 < \Pi_2$*.*

We first compare the profits in the SWOD and SWD strategies. When the seller does not provide deliberation support, the SWD strategy degenerates into SWOD strategy, i.e., $\Pi_{3s} = \Pi_{1s}$. According to Proposition 2, when $\tau < -\frac{1}{\delta} + \frac{\delta(r_s + l) + 2h}{\delta}\sqrt{\frac{c}{kh(\delta(r_s + l) + h)}}$, $\frac{\delta \Pi_{3s}}{\delta \tau} > 0$; on the contrary, when $\tau$ is larger, $\frac{\delta \Pi_{3s}}{\delta \tau} < 0$. At this time, we can get $\Pi_{3s\max} = -\frac{\sqrt{ckh(\phi\delta + h)}\left(c\delta\phi + hc - 1/2\sqrt{ckh(\phi\delta + h)}\right)}{2c\delta k(\phi\delta + h)}$, where $\phi = (r_s + l)$. For a single SaaS custom product, it is advantageous for the seller to provide deliberation support for a short period. The profit is higher than that for the product without deliberation support.

Similarly, DWD strategy degenerates into DWOD strategy, and we have $\Pi_2 = \Pi_4$ when $\tau = 0$. When the deliberation support cost of seller is low, we find that with the

increase of deliberation time $\tau$, the profit of DWD strategy is always better than that of DWOD strategy. That is, when $h < \frac{(r_p k - \phi)\delta}{2 - 2k}$, $\frac{\delta \Pi_4}{\delta \tau} > 0$ always holds. On the contrary, when the support cost is high, i.e., $h > \frac{(r_p k - \phi)\delta}{2 - 2k}$, the profit of the seller decreases first and then increases in the deliberation time, but under the high deliberation support cost $h$, the profit of DWD strategy is always lower than that of DWOD strategy.

Therefore, we recommend to sellers that when choosing the single-product strategy, it is always more advantageous to provide deliberation support than not to provide deliberation support; the key to whether to provide deliberation support when choosing the dual-product strategy is whether the deliberation support cost $h$ is low enough.

### 4.3. What Is the Seller's Optimal Product and Support Service Decisions?

In this section, we solve the optimal product strategy and we find Proposition 7.

**Proposition 7.** *When the seller's support cost satisfies $h < \frac{(r_p k - \phi)\delta}{2 - 2k}$, DWD strategy obtains the optimal profit; when $h > \frac{(r_p k - \phi)\delta}{2 - 2k}$, DWOD strategy obtains the optimal profit.*

Proposition 7 shows that the seller's optimal profit strategy depends on the seller's deliberation cost. When the seller's support cost is low, DWD strategy is the seller's optimal profit strategy; when the seller's deliberation cost is high, DWOD is the seller's optimal profit strategy.

Therefore, we recommend that sellers make decisions based on their cost of deliberation support. When deliberation support costs are low, providing deliberation support for customized SaaS products and choosing to offer both on-premises products and custom SaaS products is the seller's optimal strategy. When the seller's deliberation support cost is high, we suggest that the seller still provide deliberation support for single customized SaaS products. According to Proposition 6 and Proposition 7, we suggest that seller should provide deliberation support for customized SaaS products.

## 5. Discussion
### 5.1. The Customer Segments

We are inspired by Cheng and Liu's [15] study of the consumer segments. In the previous work [15], they constructed a consumer group matrix with trial time and customer types. They divided the consumers into four groups: free riders, buy-after-trial, buy-without-trial, and cannibalized demand consumer groups [15]. The four consumer groups reflect some real practices. For example, when the use time of software is short, some consumers can be satisfied by the free trial leading to free riders. When the use time is longer, it comes to the buyer after the trial.

However, in this study, we extend what is known in the previous study by including some emerging practices. In our work, with the observation of new practices, we get a different segment (see Figure 1) from Cheng and Liu [15]. Our results in Proposition 4 show a different finding that customers may be more willing to buy the product when the trial time is long enough. For example, in practice, during a long free trial period, consumers will record a lot of their thoughts on Evernote, and soon consumers will be inseparable from the software. Evernote becomes the king of the note-taking because it is really expensive to switch to other tools after getting used to it [26]. In this way, our work on consumer segments covers the previous findings and furthers the research of consumer group segments in the free trial field.

### 5.2. The Optimal Deliberation Time

The previous study finds that the optimal free time is no more than 1/3 of its initial version lifespan [10]. Existing empirical study also shows that the outcome of a 7-day free SaaS trial is better than the outcome of a uniform 30-day free trial [12]. Chen and Liu [15] have observed that software products need a longer trial time when products contain more

sophisticated functionalities. Here, we find that the optimal free time is related to the purchasing effort and deliberation support cost. In this work, we skillfully combine the free trial with the deliberation behavior. Proposition 2 tells that the optimal deliberation time of optimal single SaaS product strategy is $\tau^* = -\frac{1}{\delta} + \frac{\delta(r_s+l)+2h}{\delta}\sqrt{\frac{c}{kh(\delta(r_s+l)+h)}}$. We can find by numerical simulation that the optimal trial time (deliberation time) is longer when the deliberation cost is higher. In this way, we proof the observation mentioned in Cheng and Liu [15].

Figure 4 tells that the optimal free trial time is positively related to deliberation cost. If the functionality of the customized SaaS is too hard to learn, the free trial time should be long enough for consumers to learn their preference of this software. Different from the Yoganarasimhan et al. [10] who claim that 7-day trial time is better than 30-day trial time, our result is more inclusive. If the SaaS is too difficult to deliberate the preference, then a 30-day trail period may be better than a 7-day trial period.

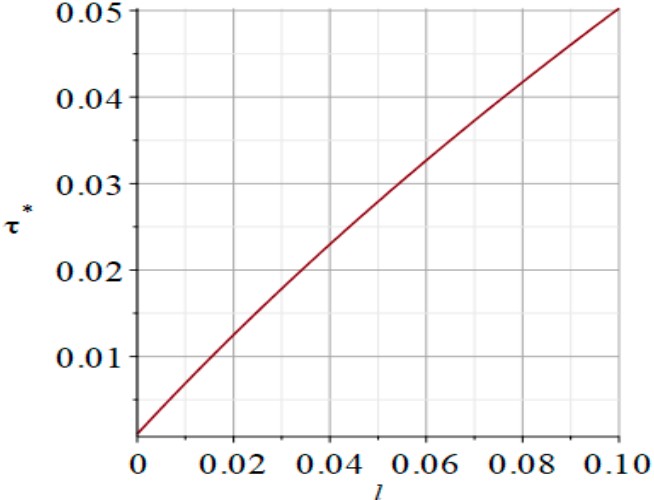

**Figure 4.** The relations between optimal deliberation time and customer deliberation cost ($\delta = 10$, $k = 0.8$; $d = 0.2$; $c = 0.1$, $r_p = 0.1$, $r_s = 0.05$, $h = 0.1$).

### 5.3. Whether to Offer Deliberation Support

Some researchers work on whether to offer free trials from the perspective of eliminating consumer uncertainty. Cheng and Liu [15] also focus on the tradeoffs between reduced uncertainty and demand cannibalization. They mentioned the double marginalization [16], demand cannibalization [15], and installed user base [2]. Our work is most related to Liu et al [14]. The previous research studied how customers' prior beliefs play a key role when software providers decide on a free trial strategy in duopoly and monopoly settings [14]. They have found that it is better to offer a free trial when the quality of substituting product is high. In our study, we build the model from the perspective of consumer deliberation behavior. The vendors offer free trials as the deliberation support responses to customers' deliberation behaviors. From Proposition 1, the vendor's profits may hurt when the consumer deliberation cost is too high. Li et al. [7] have found that retailers carrying out consumer empowerment activities can reduce consumer deliberation costs and promote channel profits. Our work is different from Liu et al. [14]: we have found that vendors who offer deliberation support can also help to promote vendor's profits by increasing consumers' WTP. Different from Li et al. [7], whether to offer deliberation support depends on the deliberation support cost. When the deliberation support cost is too high, a dual-product strategy without deliberation support would be better. But when vendors choose single SaaS strategy, it is better to offer deliberation support.

*5.4. The Contributions*

Our research mainly has the following contributions: (1) In the research field of consumer behavior, we study the scenario of the seller paying deliberation support cost for the first time. We have found that seller deliberation support costs influence sellers' product decisions. (2) Different from the previous research on SaaS in the OR field, this work considers new business practices of SaaS customization and the corresponding free trial. This consideration fills the research gap between the existing version strategies research and business practices. (3) Another novelty of this paper is the effect of a free trial on the potential consumer structure from the perspective of deliberation behavior. We reveal that the seller always needs to provide deliberation support in the optimal profit strategy, and that short-term deliberation support produces free riders while long-term deliberation support kills free riders. This finding provides some guidance for the business practices of information products.

*5.5. Implications*

We have some suggestions for sellers based on the research of deliberation support time. When the seller offers deliberation support services, a certain length of deliberation activity is always beneficial to software vendors. In the single SaaS product strategy, we have found the optimal deliberation time for vendors' reference. If the functionality of the customized SaaS is too hard to learn, the software vendors need to increase the free trial period. For those vendors who adopt the dual-product strategy, we suggest a longer deliberation time if the cost of deliberation support is low. When the cost of deliberation support is high, the deliberation support can only play a positive role for a certain period of time.

Second, the implications of consumer segments. In the single-SaaS product strategy, a short free trial will lead to free riders, which is a well-known result. If the vendor wants to increase the consumer's willingness to purchase, he/she may unexpectedly extend the trial time. Free riders disappear in case of long enough free period because of a reduced price and potential switch cost. We also remind the vendors that the deliberation support intensifies the competition of customized SaaS products over on-premises products. From Proposition 5, in dual-product strategy, when the deliberation support time is longer, the number of SaaS consumers increases while the number of the on-premises products consumers decreases. Then, we remind that seller' deliberation support increases the market share of SaaS products. We advise vendors that they should take the demand cannibalization into consideration when creating a product strategy. It is important for vendors to enrich the differentiations between the two versions to reduce the cannibalization.

Third, software vendors should think about their deliberation support costs when considering product strategies. When the seller's deliberation support cost is low, a dual-product strategy along with the deliberation support service can be used to obtain higher profits; when the seller's support cost is high, a single-SaaS product strategy and deliberation support can be used to obtain optimal profits. Especially for the customized SaaS product, we suggest that the vendor always choose to provide the deliberation support service. Although the SaaS product is entrant in the market, the customized SaaS still remains a new product for potential consumers. The deliberation service can not only help to reduce the consumers' uncertainty of the customized SaaS but also help the software providers who want to transition to the cloud service.

## 6. Conclusions

This paper studied the selection of a single-product strategy or dual-product strategy faced by the monopoly software vendors. We investigated whether and how the monopoly software vendors provide the deliberation support service for the emerging SaaS customization. In the single-product strategy, we found that the benefits of software vendors are decreasing in customization effort and consumer deliberation cost. Then, we

considered offering a free trial of SaaS as deliberation support activity to improve WTP to offset consumer deliberation cost. These are our findings.

We found that deliberation service is vital for SaaS, especially in single SaaS strategy. When the deliberation time is short, the free trial service leads to free riders in customized SaaS market. Unexpectedly, these free riders would disappear over a long enough deliberation period. While in the dual-product strategy, the deliberation support cost is the key when deciding whether to provide the services. For the optimal product strategy, DWD is the best strategy with a low deliberation support cost. When the cost of deliberation support is high, SWD is the optimal strategy.

This study still has the following limitations. This article only examines the competition generated by different products. We do not offer comparisons between software vendors on product pricing and product version strategies. In addition, we examined the impact of customizable SaaS on traditional on-premises products and ignored the common SaaS software version while the research on standard SaaS version is more common in the literature.

Future research will consider the influence of competition among software vendors on product line and consumer deliberation support. To be more specific, the future study may consider the impact of competition on the deliberation support decisions. Based on the impacts, we can determine the optimal pricing decision under the competition situations.

**Author Contributions:** Conceptualization, Z.X. and W.S.; methodology, W.S. and Q.C.; formal analysis, W.S. and R.Z.; writing—original draft preparation, W.S.; writing—review and editing, Q.C.; visualization, R.Z.; supervision, Z.X.; funding acquisition, R.Z. All authors have read and agreed to the published version of the manuscript.

**Funding:** This research was funded by [National Social Science Foundation] grant number [22AGL001] and The APC was funded by [National Social Science Foundation].

**Institutional Review Board Statement:** Not applicable.

**Informed Consent Statement:** Not applicable.

**Data Availability Statement:** This work is based on stylized economic models rather than empirical data. We use no data in this work.

**Conflicts of Interest:** The authors declare no conflict of interest.

**Appendix A**

**Proof of Proposition 1.** (a). In dual product strategy, software provider's profit is $\Pi_2 = \frac{(4d-1)\left(-kr_p^2 + 2k(r_s+l)r_p - (l+r_s)^2\right)}{2d_3} - \frac{r_p}{2}$. We can get $x_{2s}$ and $x_{2p}$ by substituting $p_{2p} p_{2s}$ and $q$ into the profit function. Then, we get $x_{2s} = -\frac{(kr_p - l - r_s)(d - 1/4)}{\sqrt{-2(d-1/4)k(k-1)\left(r_p\left(1 - r_p/2 + r_s\right)k - 1/2\left(1 + r_s\right)^2\right)}}$ and we have $(kr_p - l - r_s)(d - 1/4) < 0$ to make sure that $x_{2s} > 0$.

When we take the derivative of $r_s$, we get $\frac{\delta\Pi_2}{\delta r_s} = \frac{(kr_p - l - r_s)(4d-1)}{4\sqrt{-2k(k-1)\left(r_p\left(l - \frac{r_p}{2} + r_s\right)k - \frac{1}{2}(l+r_s)^2\right)\left(d - \frac{1}{4}\right)}}$.

Reminding that $(kr_p - l - r_s)(d - 1/4) < 0$, in this case we have $\frac{\delta\Pi_2}{\delta r_s} < 0$.

(b). $\frac{\delta\Pi_{1s}}{\delta l} < 0$, $\frac{\delta\Pi_2}{\delta l} < 0$. By the similar way, we can take the derivative of $l$, we can get $\frac{\delta\Pi_2}{\delta l} = \frac{(kr_p - l - r_s)(4d-1)}{4\sqrt{-2k(k-1)\left(r_p\left(l - \frac{r_p}{2} + r_s\right)k - \frac{1}{2}(l+r_s)^2\right)\left(d - \frac{1}{4}\right)}} < 0$ $\frac{\delta\Pi_{1s}}{\delta l} = -\frac{1}{2} + \frac{\sqrt{k(k-4c)}}{2k} < 0$. □

**Proof of Proposition 2.** The first order derivative equal to zero of profits $\Pi_{3s}$ with respect to the deliberation time $\tau$ leads to the equilibrium condition of the optimal free trial time. thus $\frac{\delta\Pi_{3s}}{\delta\tau} = 1/4\delta k(\tau\delta + 1)(l + r_s)\sqrt{-\frac{1}{(-1/4(\tau\delta+1)^2 k + c)k}} + 1/2(-l - r_s)\delta - h = 0$, then we get $\tau^* = -\frac{1}{\delta} + \frac{\delta(r_s + l) + 2h}{\delta}\sqrt{\frac{c}{kh(\delta(r_s+l)+h)}}$. Next, we check the second order derivative, and

we get $\frac{\delta^2\Pi_{3s}}{\delta\tau^2} = -4\,\frac{c\delta^2(l+r_s)}{\left(-(\tau\ \delta+1)^2k+4\ c\right)^2}\,\frac{1}{\sqrt{-\frac{1}{\left(-1/4\ (\tau\ \delta+1)^2k+c\right)k}}} < 0$. In this way, we get the optimal free time to achieve maximum profits. $\square$

**Proof of Proposition 3.** When we solve the optimal profits of case 4 and we get the seller's optimal profit is $\Pi_4 = \frac{\left(k\delta^2\tau^2+(4d-1)(k-1)\right)\left(r_p\left(2l-r_p+2r_s\right)k-(l+r_s)^2\right)}{d_6(2k-2)} + \frac{2h\tau+r_p}{2} - \frac{\delta k\tau r_p}{2k-2}$. Then we take the first order derivative with respect to $\tau$, and we can find that $\frac{\delta\Pi_4}{\delta\tau} = \frac{D}{2(k-1)(k\tau^2+4kd-4d-k+1)}$, and $D = -\sqrt{-Ak((2\ \tau^2+8d-2)k-8d+2)}\tau -B\left((2\ \tau^2+8d-2)k-8d+2\right)$, in which $A = \left(r_p\ (\phi-r_p/2)k-1/2\ \phi^2\right)$ and $B = \left((h+r_p/2)k-h-\phi/2\right)$. When $h < \frac{(r_pk-\phi)\delta}{2-2k}$ that is $B > 0$, we have $\frac{\delta\Pi_4}{\delta\tau} > 0$ always exist. When $h > \frac{(r_pk-\phi)\delta}{2-2k}$ and $\tau > \tau_\Pi = \frac{\sqrt{(-B^2k+A)(4kd-4d-k+1)}B}{-B^2k+A}$ we have $\frac{\delta\Pi_4}{\delta\tau} > 0$ otherwise $\frac{\delta\Pi_4}{\delta\tau} < 0$. $\square$

**Proof of Proposition 7.** Reminding Proposition 2, we get $\tau^* = -\frac{1}{\delta} + \frac{\delta(r_s+l)+2h}{\delta}\sqrt{\frac{c}{kh(\delta(r_s+l)+h)}}$ to make the maximum $\Pi_{3smax} = -2\frac{E(c\delta\phi+hc-E/2)}{c\delta k(\delta\phi+h)}$, $E = \sqrt{ckh((l+rs)\delta+h)}$. Then we get $\Pi_{3smax} > \Pi_{1s}$. Then we find when $h < \frac{(r_pk-\phi)\delta}{2-2k}$, $\max\{\Pi_4,\Pi_2,\Pi_{3s},\Pi_1\} = \Pi_4$, and if $h > \frac{(r_pk-\phi)\delta}{2-2k}$, $\max\{\Pi_4,\Pi_2,\Pi_{3s},\Pi_1\} = \Pi_{3s}$. $\square$

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
