# Peer review of "Optimal Software Versioning Strategy Considering Customization and Consumer Deliberation Behavior"

_jtaer, doi:10.3390/jtaer18010014_

Round 1

Reviewer 1 Report

Dear authors,

 I am pleased to review your article “Optimal software versioning strategy considering customization and consumer deliberation behavior”.

 It is a very interesting article and it was a pleasure reading it. Besides, I appreciate practical implications pointed out in your paper.

The models presented in the article are clearly described and can be replicated by anyone.

 However, it was quite difficult for me to read the "Introduction" and "Relevant literature", as you have quite a few sentences starting with [X] without indicating the author's name.

There are cases where you end a sentence with a list of bibliographic sources and then start another one directly with the source number. In my view, this makes the manuscript hard to read and hard to follow who stated what you wanted to cite.

My suggestion is to put the author's name before the source number. (e.g. row 46: "Zeithaml [23] proposes that consumers can evaluate..."

Author Response

Dear referee,

Thank you for your very careful review of our paper, and for the comments ensued. We have taken the comments to improve and clarify the manuscript. A minor revision of the paper has been carried out to take all of them into account. In the manuscript, the ‘track changes’ are included. Row numbering and page  refer to the revised manuscript are attached as a supplement to the responses.

For the comment that 'a few sentences starting with [X] without indicating the author's name', we have changed the citation format in the manuscrip.  We have highlighted the corrections in red such as ‘Hua et al [9] claim that…’. Since there are so many ‘track changes’ we refer to the revised manuscripts for review.

Reviewer 2 Report

The paper is very interesting, it is well written with adequate research methodology. I suggest following (minor) improvements:

- Figures should be larger

-The author/s should provide more details about recommendations for future research

- English language should be proofread

Author Response

Dear referee,

Thank you for your very careful review of our paper, and for the comments, corrections and suggestions that ensued. We have taken the comments to improve and clarify the manuscript. A minor revision of the paper has been carried out to take all of them into account. In the manuscript, the ‘track changes’ are included. Line numbering, page and paragraphs refer to the revised manuscript are attached as a supplement to the responses.

  • We have adjusted the figures and made them more intuitive and clear.
  • Then we have added the details about future study (highlighted in red) such as ‘Future research will consider the influence of competition among software vendors on product line and consumer deliberation support. ... Based on the impacts we can find out the optimal pricing decision under the competition situations’ (row 575-579).
  • Finally, for the English language problem, we ask native speaker for help and the ‘track changes’ are eliminated for better readability.